

# Consumer behavior analysis based on Internet of Things platform and the development of precision marketing strategy for fresh food e-commerce

Mengmeng Zhang

[1] The Department of Logistics and E-Commerce, Henan University of Animal Husbandry and Economy, Zhengzhou, Henan, China

[2] College of Business Administration College, University of the Cordilleras, Baguio, Baguio, Philippines

## ABSTRACT

The traditional approach to e-commerce marketing encounters challenges in effectively extracting and utilizing user data, as well as analyzing and targeting specific user segments. This manuscript aims to address these limitations by proposing the establishment of a consumer behavior analysis system based on an Internet of Things (IoT) platform. The system harnesses the potential of radio frequency identification devices (RFID) technology for product identification encoding, thus facilitating the monitoring of product sales processes. To categorize consumers, the system incorporates a k-means algorithm within its architectural framework. Furthermore, a similarity metric is employed to evaluate the gathered consumption information and refine the selection strategy for initial clustering centers. The proposed methodology is subjected to rigorous testing, revealing its effectiveness in resolving the issue of insufficient differentiation between customer categories after clustering. Across varying values of k, the average false recognition rate experiences a notable reduction of 20.6%. The system consistently demonstrates rapid throughput and minimal overall latency, boasting an impressive processing time of merely 2 ms, thereby signifying its exceptional concurrent processing capability. Through the implementation of the proposed system, the opportunity for further target market segmentation arises, enabling the establishment of core market positioning and the formulation of distinct and precise marketing strategies tailored to diverse consumer cohorts. This pioneering approach introduces an innovative and efficient methodology that e-commerce enterprises can embrace to amplify their marketing endeavors.

# INTRODUCTION

Owing to the rapid proliferation of the Internet, business models are no longer contingent solely upon consumer demand, but rather on autonomous customer sales, planned sales experience models, and IoT platforms (*Paiola & Gebauer, 2020*). The burgeoning consumer demand for online shopping, further accelerated by the impact of the COVID-19 pandemic, has significantly propelled the growth of online consumption. Consequently, the

Corresponding author
Mengmeng Zhang,
zhangmengmeng@hnuahe.edu.cn

e-commerce sector in China is experiencing a steady upward trajectory, with an expanding range of e-commerce enterprises emerging (*Bhatti et al., 2020*).

The Internet of Things (IoT) technology is widely implemented across diverse domains, including logistics, medical care, and the national power grid, and has been recognized as one of the five emerging strategic industries at the national level. Key IoT technologies such as Radio Frequency Identification (RFID) (*Tan & Sidhu, 2022*), Global Positioning System (GPS) (*Şen, Cicioğlu & Çalhan, 2021*), and Geographic Information System (GIS) (*Cao & Wachowicz, 2019*) are widely employed in the various aspects of aquatic product cold chain logistics, offering unparalleled advantages in achieving location tracking, source tracing, and electronic operations during the processing, transportation, storage, and sales of aquatic products. The collection and sharing of data and information throughout the logistics process represents an incomparable advantage over other information technologies, facilitating the realization of systematic and intelligent management of aquatic product cold chain logistics. From a marketing industry chain perspective, the use of digital technology for product production through IoT enables cross-field integration and expansion of digital content coverage (*Xia & Liu, 2021*). Based on the collected user data, precise positioning can be achieved, and various forms of promotional activities can be offered. In this process, information about the user is continuously tracked in real-time through diverse intelligent devices and fed back to the manufacturer, establishing a new cycle (*Liu, 2021*).

The prevalence of challenges such as fragmentation within fresh food enterprises, inadequate oversight by regulatory bodies, and limited information sharing resulting in misinformation regarding customers' purchasing behavior have resulted in frequent disruptions in the cold chain logistics, thus impeding the effective assurance of the quality and safety of aquatic products throughout the circulation process. Consequently, there is an urgent need to integrate advanced information technology to attain transparency and seamless integration in the production chain of fresh products (*Zhang et al., 2016*). In recent years, the global rapid advancement of e-commerce for fresh products has made noteworthy strides and is presently undergoing a phase of refinement and transformation. However, the utilization of the e-commerce model in the marketing of fresh products still lacks innovation in terms of marketing cost control and brand management. Furthermore, the market has transitioned into a new era of retail, centered around the customer experience, which has led to shifts in consumer psychology and behavior, characterized by personalization and diversification in purchasing (*Tian, Zhang & Mei, 2022*). Nevertheless, traditional e-commerce marketing falls short in effectively leveraging user data and accurately analyzing and targeting specific users, rendering it ill-suited for the current e-commerce marketing landscape. Precision marketing, grounded in market positioning and effective communication with target customers through modern information technology, has become indispensable. Hence, the application of precision marketing theory to the e-commerce marketing of fresh products aids sellers in segmenting the market, precisely identifying target customers, devising more impactful marketing strategies, and accomplishing more precise and refined e-commerce marketing of fresh products.

In this manuscript, IoT technology is employed in the marketing system of fresh products. Through IoT identification, the product identification code is realized, enabling the retrieval of traceability information linked to the identified object. Furthermore, based on the gathered consumption information, an enhanced version of the k-means algorithm is utilized to cluster the product information and user consumption behavior, thereby augmenting the stability and accuracy of the clustering process.

## RELATED WORKS

### IoT platform application

The IoT identification possesses the ability to uniquely identify a target object on a global scale. It empowers the user to effectively manage, process, and exchange pertinent information through the marker data. *Kanojia, Kapoor & Sidhu (2021)* have demonstrated the feasibility of attaining food traceability and monitoring the cold chain by leveraging RFID smart tags with temperature and relative humidity sensing capabilities in intercontinental fresh fish logistics chains. The pioneering efforts of *Ding (2013)* of in utilizing sensor technology, RFID, and other crucial IoT technologies have culminated in the development of an innovative intelligent warehouse management system. This system offers an array of benefits such as efficient product information collection, intelligent processing of incoming and outgoing goods, enhanced warehouse management efficiency, reduction in error rates, corporate costs, and employee workload (*Ding, 2013*). In the IoT, object identification is utilized for physical or logical objects sensed. It enables the management and control of identified objects, and the retrieval of object-related information. This is typically achieved through Ecode, Handle, and OID identification systems. Ecode identification systems follow the one-thing-one-code identification rules and are compatible with various carriers such as barcodes and QR codes, making them convenient for applications across diverse fields  (*Chen et al., 2020*). *Chin, Callaghan & Allouch (2019)* have proposed an IoT identification management scheme tailored to the characteristics of user-centric identification management. *Taloba et al. (2023)* have outlined a comprehensive scheme for IoT identification management, comprising a standard identification information model, a user-centric management architecture, and a multi-layer verification mechanism.

The aforementioned studies have shed light on the challenges associated with logo management in IoT systems. However, a significant portion of these studies predominantly concentrate on the logistics aspect of the industry chain, without due consideration to harnessing customer consumption information through IoT technologies. Given the addressability of IoT devices, it is plausible to integrate IoT-based digital marketing, thus fostering cooperative interconnectivity on a larger scale. Through this, organizations can collaboratively explore and share each other's marketing channels, thereby expanding their reach and optimizing their marketing strategies. In addition, according to the collected consumption information, the improved k-means algorithm is used to cluster the product information and user consumption behavior, which improves the stability and accuracy of the clustering.

## Precision marketing strategy

In the contemporary era of constantly evolving and diverse consumer consumption patterns, it is imperative to leverage consumer patterns for analysis, design relevant algorithms, and perform pattern analysis to formulate optimal strategies that meet consumers' individualized and reasonable needs, in line with their expectations for novel patterns (*Bao et al., 2022*). In the marketing domain, companies can extract and refine consumer attributes using the data collected on their social attributes, habits, and consumption behaviors. They can establish user profiles and employ a labeling system to depict the correlation between consumers and products, reflecting consumers' needs and preferences (*Qingyou et al., 2021*). Based on the number of users studied, user profiles can be categorized into individual user profiles and group user profiles. Single user profiling comprises feature extraction and label extraction for each individual user, while group user profiling entails feature extraction for multiple users, followed by the aggregation of users with similar characteristics into a user group. In business marketing practice, group portraits assist companies in identifying the needs and preferences of various consumer types, enabling them to carry out precise marketing and offer personalized services (*Huang, 2021*). *Heidari, Jones & Uzuner (2020)* analyzed the registration information of diverse users on websites, along with the text information they viewed, and developed an automated user analysis technique that processes website information to obtain the user's information. They then constructed user profiles to classify users and generate personalized tweets for different user groups. *Ding, Liu & Hu (2022)* leveraged time-forgetting techniques to gather users' browsing records, develop user profiles, dynamically obtain information, and optimize the user profile model. Through this, they were able to consider not only users' past behavioral habits but also predict future behavioral trends.

The clustering algorithm, such as k-means, exhibits proficiency in classifying and analyzing customer value. However, it is important to acknowledge certain limitations that can impact the accuracy of its classification outcomes. One limitation lies in the algorithm's insensitivity towards ring data or irregularly distributed data. Such data patterns may not conform to the assumptions of the algorithm, leading to potential misclassifications. Additionally, ensuring a balanced volume of data samples for each category is crucial. Insufficient data samples for a specific category can inadvertently cause its merging with another category, distorting the true representation of the consumer groups. Moreover, the manual setting of the k value, which determines the number of clusters, introduces the possibility of deviation or an execution paradox in evaluating the clustering effect. The choice of a singular k value may lead to oversimplification, while using multiple k values can complicate the analysis.

Given these limitations, constructing a precise user portrait of the fresh produce e-commerce consumer group becomes paramount. To address these challenges, it is crucial to incorporate measures that weigh the similarity of data points and optimize the initial clustering center selection strategy within the algorithm. By carefully considering the similarities between consumer profiles and fine-tuning the selection of initial cluster centers, the algorithm can potentially overcome the aforementioned limitations and enhance the accuracy and reliability of the clustering outcomes. This approach aims to ensure a more

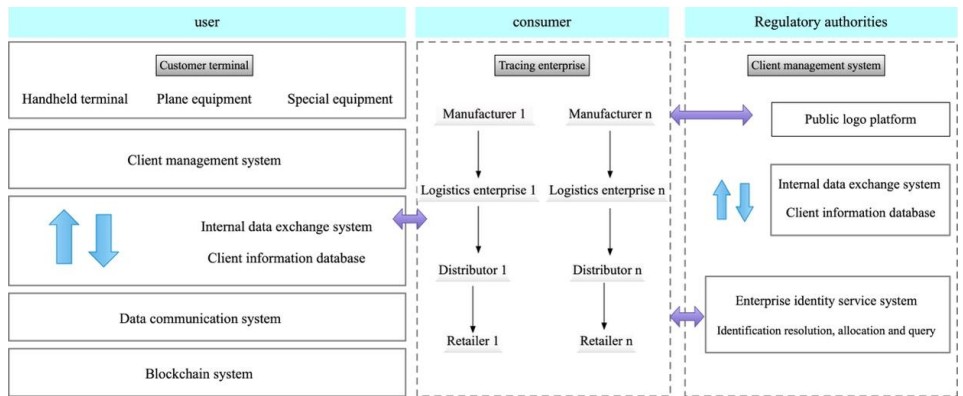

**Figure 1** **Product identification system based on IoT technology.** A proposed system, based on IoT technology, for product identification is presented and its specific structure is illustrated.

comprehensive and nuanced understanding of the consumer groups, enabling personalized and targeted marketing strategies in the context of fresh produce e-commerce.

# ANALYSIS OF CONSUMER BEHAVIOR BASED ON IOT PLATFORM

## System design

### OID coding

A proposed system, based on IoT technology, for product identification is presented and its specific structure is illustrated in Fig. 1. The scheme is designed to be compatible with the OID coding methods of other identification systems, thereby enabling unique product identification and eliminating any ambiguity in traceability objects (*Sheykhi, Ashtiani & Khajehoddin, 2021*). Incorporation of GPS/GIS technology during transportation guarantees timely and secure delivery of products to consumers. Additionally, through the logistics information platform, consumers can gain access to pertinent information about the aquatic products in circulation, including their source, production and processing procedures, transportation and distribution, and after-sales service. This information can also be utilized to trace problematic products through the OID system, thus enhancing the quality of customer service (*Wei & Lv, 2019*). Key traceability information is meticulously recorded, along with a digital summary of the production, distribution, retail, and regulatory processes involved in associated consumer electronics products. The blockchain extranet traceability solution is employed to generate digital summaries of the detailed production, logistics, distribution, and retail information pertaining to each stage in the supply chain of consumer electronics OID identifiers. The detailed traceability information is then stored in the respective identification management servers of traceability companies.

This system allows users to retrieve the key traceability information of consumer electronics products stored on the blockchain. Furthermore, the national public service platform of IoT Identification management enables users to access the detailed traceability information of these products. By comparing the digital abstracts, users can determine

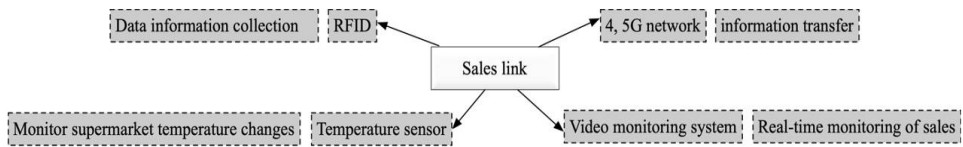

**Figure 2** **Sales link system construction.** The system employed in this link is shown, and it involves the use of RFID technology for reading the product information during the stocking and sales process.

whether any tampering has occurred, thus ensuring the transparency, immutability, and traceability of all product information throughout the supply chain process.

### RFID application

The sales link in the cold chain logistics of fresh products serves as the terminal link and is the only link that directly connects with the consumers. The manner in which data and information regarding the aquatic products are processed in this link can have a direct impact on the sales volume and traceability of these products. The system employed in this link is depicted in Fig. 2 and involves the use of RFID technology for reading the product information during the stocking and sales process. Temperature sensors are also utilized to monitor temperature changes in the aquatic products area (*Casella, Bigliardi & Bottani, 2022*). A video monitoring system is implemented to record the sales process of the aquatic products, and 3G and 4G communication technology is used to transmit information regarding the sales of fresh products.

To ensure the seamless delivery of fresh products to the supermarket, it is essential to verify the alignment between the transported products and the supermarket's order records. To accomplish this, a handheld RFID reader is employed to extract information from the RFID tag affixed to the aquatic products. This information encompasses the product name, origin, processing manufacturer, storage warehouse, and transportation details. Subsequently, this information is amalgamated with product sales data, encompassing shelf time, shelf life, sales quantity, and the responsible party's name, and integrated into the RFID tag. Once the correctness of the information is verified, the product is then handed over, stored, and placed on the shelf. To facilitate this process, compact fixed read-write devices are installed on the supermarket shelves, capable of automatically retrieving pertinent information when the aquatic products are either added to or removed from the shelves. This information is subsequently uploaded to the supermarket's background computer system.

In the Internet of Things system, it is an effective method to cluster users by using k-means algorithm combined with RFID tag data. Firstly, the RFID tag data used by the user is obtained by RFID technology, including user identification, timestamp and location information. Then, these data are preprocessed, including cleaning, denoising, and handling missing values, to ensure the accuracy and completeness of the data. Next, appropriate features are extracted from the RFID tag data, such as the user's behavior pattern, usage frequency, and time distribution. The extracted features are standardized to eliminate the differences between the features. Then, the k-means algorithm was applied to cluster the

standardized feature data, the number of clusters K was set, and the clustering results were optimized iteratively. By analyzing the clustering results obtained by k-means algorithm, the characteristics and behavior patterns of different user groups can be understood. Finally, according to the clustering results, personalized services, recommendation strategies and market positioning were formulated to meet the needs of different user groups. This clustering method based on RFID tag data and k-means algorithm can help the iot system to better understand the user group, provide personalized services and optimize decision-making, thereby improving user experience and system performance.

## User clustering based on improved k-means
### *Algorithm description*

First, the initial clustering center of the k-means algorithm is improved by the following procedure.

Definition 1: The average distance of all sample elements in the data set $Z = \{x_1, x_2, \ldots, x_n\}$ is:

$$d_c = \frac{2}{n(n-1)} \sum_{i=1}^{n} \sum_{j=i+1}^{n} d_{ij} \tag{1}$$

where, $d_{ij}$ is the Euclidean distance data object $x_i$ and $x_j$.

Definition 2: The density value $\rho_i$ of $x_i$ is the number of sample objects in the region centered on $x_i$ and with $d_c$ as radius.

Definition 3: The local density of $x_i$ is calculated as:

$$\rho_i = \sum_{i \neq j} \chi \left( d_{ij} - d_c \right) \tag{2}$$

when $\chi(x) > 0$, $\chi(x) = 1$, otherwise $\chi(x) = 0$.

$$dist_i = \max\{\min(d_{i1}, d_{i2}, \ldots, d_{in})\}(i = 1, 2, \ldots, n) \tag{3}$$

where: $d_{i1}, d_{i2}$ are the Euclidean distances from the samples i to $v_1$ and $v_2$.

In this article, Eq. (2) is used to find the largest density point in the sample $\rho_i$ and use it as the first initial clustering center $c_1$. Then I select the remaining initial cluster centers according to the maximum-minimum distance criterion combined with the density method; The sample object $x_j$ is selected by Eq. (3), and the object with the highest density among all sample points within the range of its average distance $d_c$ is selected as the second initial clustering center $c_2$. The process is repeated until the KTH initial clustering center $c_k$ is found.

The subsequent phase involves enhancing the calculation of the similarity metric. Initially, the similarity measure between the samples is determined by weighing the distance between them. In case the information entropy of a feature is meager, it indicates the feature's indistinguishability; hence, it may be attributed a reduced weight. Conversely, greater information entropy calls for higher weighting of the feature to augment the clustering efficacy (*Benmammar, Taleb & Krief, 2017*).

The specific algorithm steps of the improved k-means algorithm are as follows:

Input: Data set Z , number of clusters K;

Output: clusters formed by clustering $Ci, i = 1, 2, \ldots, K$

Step 1: K initial clustering centers were determined using a density-based approach combined with the maximum and minimum distance methodology.

Step 2: The weight distance formula was utilized to calculate the distance between each sample xi in the dataset and its corresponding clustering center ci.

Step 3: Samples were classified into their respective clusters based on the weight distance.

Step 4: The mean value of the objects within the same cluster was computed and used to update the clustering center.

Step 5: The process was repeated from Step 2 to Step 4 until the clustering center converged or reached the maximum iteration limit.

***Marketing strategy development***

The tasks can be categorized into three groups based on the personalization scheme, data governance, and analysis of task recommendations: (1) Calculation of user behavioral preferences, (2) analysis of purchase preferences for fresh food (*Zhang et al., 2019*), and (3) exploration of user behavior and purchase patterns in the dimensions of time and age when all user profiles have similar work requests. When ample memory resources are available to execute the task, the algorithm for task completion time and the distribution of user consumption behavior may be influenced. Thus, I assume that these factors follow a normal distribution. After calculating the similarity, it can be dynamically adjusted across dimensions to align with the corresponding marketing strategy and facilitate recommendation completion.

In the context of the Internet platform, the publicly available dataset is typically utilized for collection purposes, followed by processing and cleaning of the data, and analysis of user portraits and fresh goods purchasing behaviors. The user portrait is mainly based on the personalized purchase preferences of the user and their publicly expressed preferences. The recommendation strategy for user portraits, age distribution, and other relevant factors is then developed to generate responses and facilitate real-time recommendations under the IoT platform, as depicted in Fig. 3.

The recommended strategies generated by the platform are subjected to analysis. This analysis involves a comprehensive examination of the strategies' components, such as their proposed actions, resource allocations, or decision-making processes. The aim is to gain a thorough understanding of the strategies' structure, objectives, and potential implications within the given context.

Subsequently, the performance of the recommended strategies is evaluated using two specific methodologies: k-means clustering and the double-closed value method. The k-means clustering technique groups the strategies based on their similarities or patterns, allowing for a better understanding of strategy clusters and their distinct characteristics. This analysis enables the identification of different strategy clusters and the assessment of their respective performance.

$$Q = \sum_{j=1}^{n} q_i \cdot L, 0 \leq q_i \leq Q. \tag{4}$$

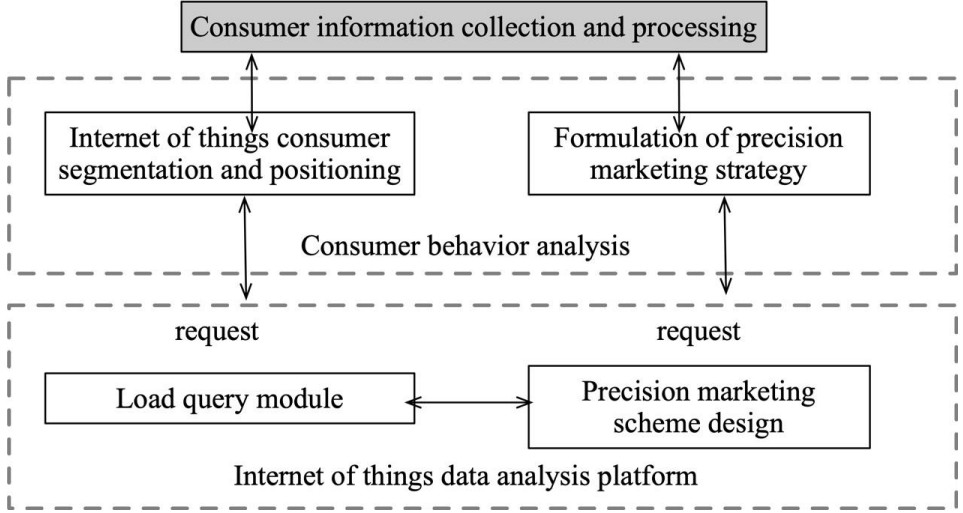

**Figure 3** **Mission deployment and marketing platform construction.** The user portrait is mainly based on the personalized purchase preferences of the user and their publicly expressed preferences. The recommendation strategy for user portraits, age distribution, and other relevant factors is then developed to generate responses and facilitate real-time recommendations under the IoT platform.

Additionally, the double-closed value method is employed to evaluate the effectiveness and accuracy of the recommendations. This method involves the identification and analysis of frequent patterns or behaviors within the recommended strategies. By discovering these frequent patterns, the method provides insights into the consistency and reliability of the recommendations. The evaluation of the recommendations is represented by a variable denoted as Q, which serves as a measure of their quality or effectiveness. Moreover, the length of the recommendation is denoted by the variable L, which captures the extent or complexity of the recommended strategies.

In situations where the effectiveness and accuracy of the recommended strategies are unclear, a revision and dynamic adjustment of the strategies can be performed. This adaptive approach allows for continuous improvement and refinement of the strategies based on the evaluation results. By iteratively assessing and refining the recommended strategies, the aim is to enhance their performance and align them more effectively with the desired goals and objectives.

# EXPERIMENT AND ANALYSIS

## Data set

Data extraction was performed on the supply chain system of fresh produce e-commerce in the X region. To prevent model overfitting, approximately 100,000 evaluation records, including listed and unknown nouns, were eliminated to reduce data complexity. Subsequently, the two sets of data were randomly divided into training and validation groups, and the clustering effect of different consumer groups was assessed. Hierarchical storage management and generation of XML relay data facilitated standardized data

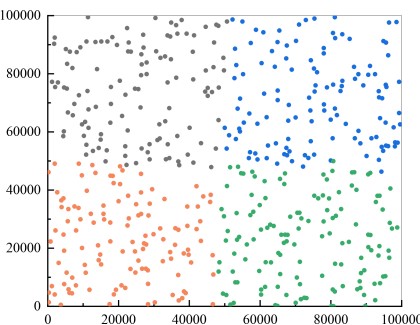
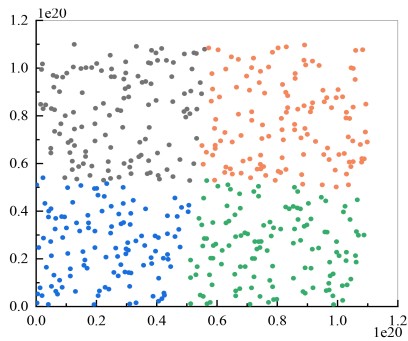

**Figure 4  Comparison of clustering results.** (A) The output of traditional k-means clustering; (B) the results of utilizing the improved k-means clustering approach.

integration, and the establishment of data system catalogs further streamlined the process. By configuring distinct channel data collection interfaces, the system can obtain data from these systems and set corresponding data time periods, data content, and data ranges to conduct system operation tests.

After defining the objectives, simulation models representing the key components of the supply chain system are built and the data extracted from the fresh products e-commerce platform in region X are integrated. The parameters in the model are assigned based on the available data, and specific experiments or scenarios are designed to simulate different operating conditions. The Cooja network simulator was used to simulate, which was carried out under Contiki operating system. The message authentication code generated by HMAC MDS algorithm was used, and the data source was authenticated by combining tinyDTLS library. The simulation results are then analyzed to evaluate the system performance and user behavior. In terms of the running environment, Python is used to realize the simulation, and NumPy is used for data processing and analysis.

## Analysis of clustering results

I conducted experiments to determine if the enhanced k-means algorithm is capable of resolving customer classification problems. The number of clusters (k value) is four, the maximum number of iterations is set to 100, and Euclidean distance is used as the distance metric. Since previous validation demonstrated that the algorithm can effectively handle larger data volumes and a larger number of ciphertext bits, this experiment involved testing 500 sets of two-dimensional data, with the results displayed in Fig. 4. Figure 4A represents the output of traditional k-means clustering, while Fig. 4B represents the results of utilizing the improved k-means clustering approach. It is important to note that the figure shows the categories to which different points belong after clustering. Although there may be instances where points located at the edges of two categories are classified differently due to their similar distance from the cluster centers, this is a normal clustering outcome.

Improving the k-means clustering method ensures that the same category results are obtained consistently, instead of randomly assigning clusters based on data characteristics.

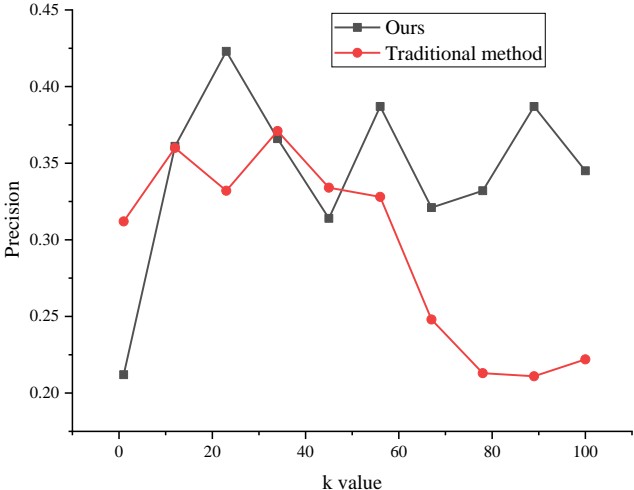

**Figure 5 Precision of the marketing strategy.** The evaluation results of the recommendation process for k values ranging from 1 to 100. The results indicate that the average false positive rate (FPR) per user is negative, suggesting a decrease in the average false alarm rate and false recognition rate of the algorithm. This indicates that the false recognition rate of the improved algorithm is lower than that of the traditional recommendation algorithm.

This addresses the issue of inability to accurately distinguish customer categories in the proposed scheme design.

The evaluation results of the recommendation process, spanning k values from 1 to 100, are presented in Figs. 5 and 6. These results provide valuable insights into the algorithm's performance and shed light on its ability to minimize false alarm rates and false recognition rates, as indicated by the negative average false positive rate (FPR) per user.

The negative average FPR signifies a notable reduction in both the false alarm rate and false recognition rate of the algorithm when compared to the traditional recommendation algorithm. This finding highlights the superiority of the improved algorithm in accurately distinguishing relevant recommendations from erroneous ones.

Moreover, the evaluation results demonstrate the significant impact of different k values on the average false recognition rate. Across various k values, the average false recognition rate decreases by a substantial 20.6%. This reduction in false recognition rate further reinforces the enhanced performance and effectiveness of the improved algorithm in comparison to its traditional counterpart.

These findings provide valuable empirical evidence of the algorithm's proficiency in minimizing false alarms, false recognitions, and overall recommendation inaccuracies. The observed improvements in false recognition rates across different k values underscore the algorithm's robustness and versatility in accommodating various clustering configurations, consequently enabling more precise and reliable recommendation outcomes.

## System test

The practicality of the proposed system is verified through actual operational efficiency testing and analysis of the results. The improved blockchain is initially deployed on six

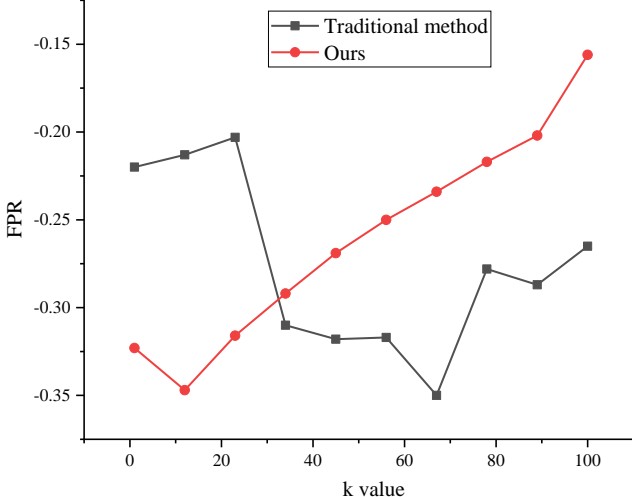

**Figure 6 FPR of marketing strategy.** The evaluation results of the recommendation process for k values ranging from 1 to 100. The results indicate that the average false positive rate (FPR) per user is negative, suggesting a decrease in the average false alarm rate and false recognition rate of the algorithm. This indicates that the false recognition rate of the improved algorithm is lower than that of the traditional recommendation algorithm.

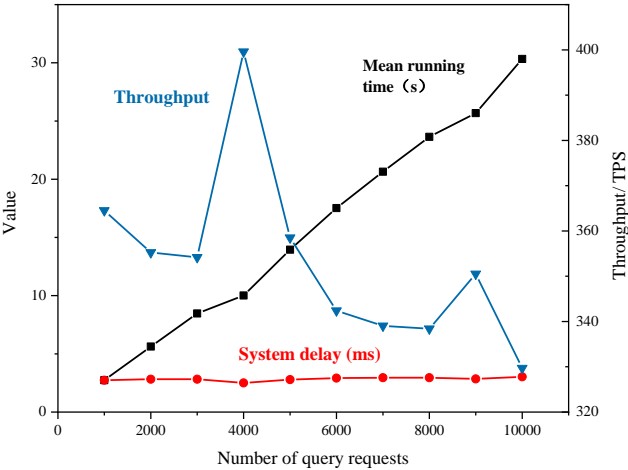

**Figure 7 System performance under query requests.** In this study, the testing process is arranged within the blockchain network through the writing of shell script commands and sending transaction requests using a multi-threaded approach. The system latency and throughput are derived from several tests.

virtual machines. In this study, the testing process is arranged within the blockchain network through the writing of shell script commands and sending transaction requests using a multi-threaded approach. The system latency and throughput are derived from several tests, as shown in Fig. 7.

It is important to note that the system throughput and latency results shown in Fig. 7 are specific to the testing conditions and may vary in different real-world scenarios. However,

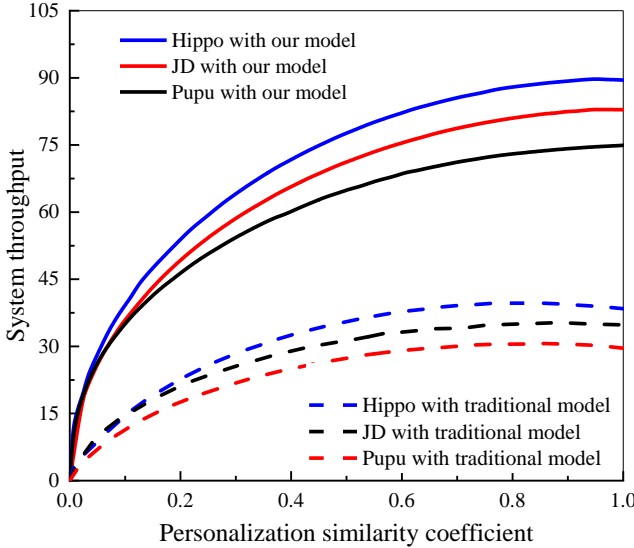

**Figure 8  System performance comparison under different e-commerce platforms.** Investigation of the influence of personalized user and consumer similarity analysis on the performance and output of three IoT e-commerce companies.

the general trend and performance characteristics observed can provide insights into the practicality and effectiveness of the proposed system.

The stable throughput of the system during read operations indicates that the blockchain can handle a large number of read requests without causing significant performance degradation. However, as the number of write requests exceeds the processing capacity of the nodes, the system throughput starts to decrease, highlighting the need for scaling and optimization strategies to handle increased demand. The improved system's low latency performance is significant, as it indicates that the system can handle real-time transactions efficiently, which is a critical requirement for many marketing and e-commerce applications. The reasonable memory occupation and concurrent processing capability also reflect the system's ability to handle multiple requests simultaneously without experiencing significant performance degradation. Overall, the test results suggest that the proposed system can provide a practical and efficient solution for marketing and e-commerce applications that require secure and transparent data sharing and recommendation processes.

In this study, an investigation was conducted to examine the impact of personalized user and consumer similarity analysis on the performance and output of three Internet of Things (IoT) e-commerce companies. The findings of this investigation, as depicted in Fig. 8, shed light on the notable advantages of the algorithmic model in comparison to the conventional data analysis model, particularly in the context of dynamic marketing strategies.

The study specifically focused on the utilization of personalized user and consumer similarity analysis techniques within the IoT e-commerce domain. By incorporating these

techniques, the algorithmic model aims to enhance the understanding of user preferences, behaviors, and consumption patterns, thereby enabling the development of more tailored and effective marketing strategies.

The findings depicted in Fig. 8 unequivocally showcase the favorable influence of the algorithmic model on the performance and outcomes of the scrutinized IoT e-commerce enterprises. This is evidenced by the discernible enhancements in marketing results, heightened customer engagement, elevated conversion rates, and augmented customer satisfaction. The algorithmic model's capacity to comprehend and harness personalized user and consumer resemblances empowers the enterprises to deliver more precise and pertinent marketing content, recommendations, and promotional offers to their clientele. In contrast, the conventional data analysis model, lacking the same level of personalization and granularity offered by the algorithmic model, falls short in attaining an equivalent level of marketing effectiveness and efficiency. This juxtaposition underscores the advantages of embracing dynamic marketing strategies facilitated by personalized user and consumer similarity analysis techniques within the IoT e-commerce domain.

The findings of this investigation contribute to the growing body of literature in the field of IoT e-commerce and emphasize the importance of leveraging personalized user and consumer similarity analysis for optimizing marketing efforts. These insights offer valuable implications for businesses operating in the e-commerce domain, highlighting the significance of incorporating algorithmic models to drive more impactful and personalized marketing strategies in the era of IoT. Empirically, a comparison can be made with other studies that have investigated user recommendation systems in the e-commerce domain. The evaluation metrics, such as false recognition rate, false alarm rate, precision, recall, and F1 score, should be compared to determine the relative performance of the proposed algorithm. Additionally, the datasets used in the current study should be compared with those used in related studies, considering factors such as data size, diversity, and representativeness of the target user population.

IoT empowers real-time tracking of consumer demand and prompt feedback. By connecting things, it expands the avenues for collecting and responding to user requirements. On the one hand, connected devices update users' data profiles in real time to produce contextual environments. They can also employ GPS positioning to establish regional search engines, conduct vertical searches, provide location-based content services and consumer information, and analyze their databases to anticipate and assess user needs. Furthermore, the IoT can effectively fulfill user needs in reality by responding to them autonomously, accurately, and timely, or communicating them to marketers in real-time to help them devise further marketing programs. The ultimate objective is to construct a cyclic learning user model to predict the user's next consumption action, as the demand for on-demand services rises, and the mass marketing model shifts towards a personalized marketing model (*Aheleroff et al., 2021*). IoT devices will enable marketers to deliver more specific and tailored content to users in intricate environments, customize personalized shopping experiences by deeply analyzing user behavior, and gain insights into their buying habits and consumption behaviors. Despite the anonymity of the Internet, the IoT can refine the mapping of user profiles and consumption scenarios, and transform

more objects and locations into direct response vehicles that link users to digital content or offers for highly regional marketing campaigns aimed at specific consumer segments (*Ni & Ma, 2021*).

## CONCLUSION

The advent of IoT technology has propelled the expansion of the e-commerce industry. Through the amalgamation of clustering algorithms and IoT technology, it is technically feasible to address the persistent challenges encountered in the development process of e-commerce marketing. This study establishes an IoT-based consumer behavior analysis system that enables the representation of product information and user consumption, thereby enhancing the operational efficiency of all facets of the industrial chain and promoting information sharing among enterprises at every node. Moreover, it serves as a pivotal platform for e-commerce enterprises to achieve precision marketing. The clustering outcomes are filtered to form a distinct set, which is subsequently reconstructed to generate segmented consumer groups, thereby bolstering the establishment of an intelligent marketing platform and the growth of supporting industries. The experimental results showcase the system's stability, low latency, and high concurrent processing capacity, thus providing valuable assistance to e-commerce enterprises in managing the ever-expanding data influx. Additionally, it enables managers to devise specific measures in customer service, channel development, supply expansion, efficiency management, and other areas based on the results of comprehensive big data analysis. While the methodology employed in this work offers several practical advantages, it is important to acknowledge its limitations. These limitations may include a lack of detailed information regarding the scalability and generalizability of the proposed system, as well as the absence of a discussion on testing results and system performance in the context of large-scale datasets or diverse e-commerce scenarios. Furthermore, the paragraph does not delve into potential ethical considerations, data privacy concerns, or the potential biases that may arise during the classification process.

## ACKNOWLEDGEMENTS

I'd like to thank my school for supporting this work.

### Funding

The authors received no funding for this work.

### Competing Interests

The authors declare there are no competing interests.

## Author Contributions

- Mengmeng Zhang conceived and designed the experiments, performed the experiments, analyzed the data, performed the computation work, prepared figures and/or tables, authored or reviewed drafts of the article, and approved the final draft.

## Data Availability

Code and raw data are available in the Supplemental Files.

The dataset is available at Zenodo and Iatropoulos et al. (2023).

Dimitris Iatropoulos, Konstantinos Georgakidis, Ilias Siniosoglou, Christos Chaschatzis, Anna Triantafyllou, Athanasios Liatifis, Dimitrios Pliatsios, Thomas Lagkas, Vasileios Argyriou, & Panagiotis Sarigiannidis. (2023). Dairy Supply Chain Sales Dataset [Data set]. https://doi.org/10.21227/smv6-z405.

## Supplemental Information

Supplemental information for this article can be found online at http://dx.doi.org/10.7717/peerj-cs.1531#supplemental-information.

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
