# Peer review of "Consumer behavior analysis based on Internet of Things platform and the development of precision marketing strategy for fresh food e-commerce"

_PeerJ Computer Science, doi:10.7717/peerj-cs.1531_

## Round 0.1 · original submission · Major Revisions

Dear authors
Thanks for your submission, you will see that they have couple of comments which needs your attention, I do also agree with the experts and also the following additional improvements needed while resubmitting:
1. Please improve the abstract to include the problem statement, your concise solution, the methodology adopted and the obtained results.
2. Please carefully revise the technical language of the paper.
3. Clearly write the conclusion and possible limitations
Thanks

Reviewer 1 ·

Basic reporting

The K-means algorithm is embedded in the designed system to classify consumers. According to the collected consumption information, the similarity measure is used to do weighting processing, and the selection strategy of the initial clustering center is optimized. The test results show that the proposed method effectively solves the problem that the category of customers cannot be distinguished after clustering. It is helpful to further segment the target market and core market positioning, carry out different precision marketing strategies for different consumer groups, and provide a new method that can be used for reference.
This is a valuable and interesting idea. However, with the current quality, this article cannot be published. This article has many defects, so my suggestion is a minor revision.

Experimental design

A number of comments listed below:
Add further details on how simulations were conducted. Resource and system characteristics could be added to Tables for clarity. The paper lacks the running environment, including hardware and software details.
The motivation for using many different methods is not described properly. All of these combined methods need the appropriate values for their parameter. Robustness should be discussed;
Internet of Things technology is a very broad concept, but from the author's description, this research only uses RFID, so some organizational structures are unreasonable;

Validity of the findings

A number of comments listed below:
What role does Section 3.1.2 play in this study? From the analysis of the results, the introduction of the sales system seems redundant;

The author uses the strategy recommended by the platform for analysis, and then uses K-means and double-closed value method to evaluate the state of the strategy. Formula (4) needs more discussion;
There is a lack of comparison with other methods. The authors should compare their work with more effective methods or explain the current comparison in detail.It could also be used to investigate the system without the initial assumption and to compare the performance results with and without those assumptions.
The values for the parameters of the algorithms selected for comparison are not given.

Additional comments

The authors should clarify the pros and cons of the methods. What are the limitation(s) methodology(ies) adopted in this work? Please indicate practical advantages, and discuss research limitations.

Reviewer 2 ·

Basic reporting

The rise of Internet of things technology has promoted the development of e-commerce industry. Combining clustering algorithm and Internet of things technology to solve the problems existing in the development process of e-commerce marketing is feasible in technology.

The experimental results show that the stability, low latency and high concurrent processing ability of the design system can help e-commerce enterprises face the expanding data explosion and assist managers to put forward specific measures in customer service, channel development, supply and sales expansion, and efficiency management based on the results of big data analysis.

Experimental design

(1) To have an unbiased view in the paper, discuss the limitations of your study. These limitations can be organized around simple distinctions of the choices you made in your study.
(2) There are some statements missing in references 15-19, and the author needs to add a new paragraph to review these citations.
(3) Further explain the logical relationship between the headings, they need to be arranged in a more reasonable way.
(4) Discuss your position on the generalizability of your results. Clarifying the study’s limitations allows the readers to better understand under which conditions the results should be interpreted.
(5) The analysis and configurations of experiments should be presented in detail for reproducibility. It is convenient for other researchers to redo your experiments, and this makes your work easy acceptance. A table with parameter setting for experimental results and analysis should be included in order to clearly describe them.

Validity of the findings

(6) The experimental results are poor. Different metrics should be used for simulation results.
(7) Are the simulation results taken from the equal conditions? There is not any discussion.
(8) The paper should be revised by adding a more extensive methodological and empirical comparison with closely related papers, so that the significance of the proposed method becomes evident.
(9) A clear description of limitations of a study also shows that the researcher has a holistic understanding of his/her study. However, the authors fail to demonstrate this in their paper.

---

## Round 0.2 · accepted · Accept

Based on the input of experts on the revised version of the paper, I'm pleased to recommend it for publication. Thanks for your consideration of this journal.

Reviewer 1 ·

Basic reporting

All comments have been addressed by the author

Experimental design

All comments have been addressed by the author

Validity of the findings

All comments have been addressed by the author

Additional comments

NA

Reviewer 2 ·

Basic reporting

The suggested comments are now well explained in the revised version.

Experimental design

In the revised version, the comments are positively incorporated.

Validity of the findings

No Comments